



# Spatiotemporal variation in rainfall predictability in Serbia under a changing climate

Tatijana Stosic[1], Ivana Tošić[2], Antonio Samuel Alves da Silva[1], Vladimir Djurdjević[2], Borko Stosic[1]

[1]Department of Statistics and Informatics, Federal Rural University of Pernambuco, 52171-900 Recife-PE, Brazil

[2]Faculty of Physics, Institute for Meteorology, University of Belgrade, Belgrade, 11000, Serbia

*Correspondence to*: Antonio Samuel Alves da Silva (antonio.sasilva@ufrpe.br)

**Abstract.** This study examines whether the predictability of precipitation dynamics in Serbia has been influenced by climate change. We apply Generalized Weighted Permutation Entropy (GWPE) to evaluate the temporal structure of daily precipitation series using the parameter $q$, which filters subsets of small ($q < 0$) and large ($q > 0$) fluctuations. The analysis covers data from

14 weather stations between 1961 and 2020.

Entropy values for $q = 0$ and $q = 2$, corresponding to Permutation Entropy and Weighted Permutation Entropy respectively, remained stable spatially and temporally. In contrast, GWPE values for $q = -10$ and $q = 10$, representing the predictability of small and large fluctuations, exhibited significant spatial and temporal variation between two 30-year subperiods. Entropy values for $q = -10$ were consistently lower, indicating that small precipitation fluctuations are more predictable than large ones.

In several locations, significant changes in entropy occurred despite relatively stable annual precipitation amounts. In others, annual totals varied while entropy remained constant. These findings suggest that climate change has influenced the predictability of precipitation in Serbia. By filtering fluctuations across scales, GWPE effectively reveals underlying changes that may be masked by standard statistical measures.

## 1 Introduction

The complexity of climate systems is widely recognized by researchers from different areas, such as physics, hydrology and ecology (Rind, 1999; Mihailović et al., 2014; Dey and Mujumdar, 2022; Almeida-Ñauñay et al., 2021), and the research that addresses this aspect of climate has been intensifying over the recent years. In particular, there has been an increasing number of publications that address the complexity of climate variables by using the methods developed in complex system science, such as fractal and multifractal analysis (Lovejoy and Mandelbrot, 1985; Krzyszczak et al., 2019), methods originated in

information theory (Silva et al., 2021a), and more recently complex networks (Boers et al., 2019). Precipitation and temperature are used to describe the climate of certain regions (Da Silva et al., 2019), for climate classification (Peel et al., 2007), and to detect possible climate change (Alexander et al., 2006).

Climate models are widely used to predict future climate conditions, but must be validated against historical observations for different temporal and spatial scales to evaluate the models' performance for future projections (Vautard et al., 2021).



Empirical results indicate that climate models perform better in reproducing temporal and spatial patterns of surface temperature than those of precipitation (Trenberth, 2011; Lagos-Zuniga et al., 2022), thus other aspects such as emergent properties that are characteristics of complex systems could contribute to better understanding of the nature of stochastic processes that govern spatial and temporal precipitation variability, and help in evaluation of performance of climate models and the impact of climate change. Fractal, multifractal, and complex network analyses have been extensively utilized to explore

the temporal and spatial variability of rainfall, while the potential of information theory-based methods in uncovering aspects of rainfall dynamics remains less explored. Recently, entropy measures have been applied to characterize irregularities and the rate of information flow in hydrological data, providing insights into different regimes and the effects of natural and human-induced factors (de Carvalho Barreto et al., 2020; Agarwal et al., 2016; Rolim and de Souza Filho, 2025).

However, these entropy measures lack the ability to capture temporal relationships between successive values in a time series,

which was incorporated in the Permutation Entropy method introduced by Bandt and Pompe (2002). Rosso et al. (2007) combined Permutation Entropy with a statistical complexity measure to develop the "Complexity-Entropy Causality Plane" (CECP). This tool has proven effective in comparing systems with different degrees of complexity, especially in differentiating between noise and chaos.

In hydrology Permutation entropy methods were used to analyze the nature of stochastic process governing streamflow

dynamics (Serinaldi et al., 2014; Mihailović et al., 2024), to optimize streamflow monitoring networks (Stosic et al., 2017), to improve prediction models (Wang et al., 2024) and for detecting hydrological alterations caused by natural and human factors (de Carvalho Barreto et al., 2023; Suriano et al., 2024). For a comprehensive review of the use of Permutation entropy-based methods in hydrological studies see a recent article of Mihailović (2025). On the other hand, there are only a few studies that use CECP to investigate the complexity of rainfall. Silva et al. (2021b) applied CECP method on monthly rainfall data recorded

in 133 pluviometric stations in Pernambuco, Northeast Brazil, during the period from 1950 to 2012, and showed the potential of CECP to distinguish between different rainfall regimes. Tongal (2025) used CECP to improve the detection of hydrological communities in the Lake District, Türkiye. Du et al. (2022) used PE along with other entropy measures to examine the rainfall complexity in Beijing, China. Based on the monthly precipitation data recorded from 58 meteorological stations between 1968 and 2017, they observed significant spatial variations in rainfall complexity across the study area. They also found that rainfall

complexity has been increasing due to both climate variability and human activities. Liu et al. (2020) applied multiscale permutation entropy to analyze precipitation complexity across 13 major cities in Heilongjiang Province, China. Based on monthly data from 1967 to 2016, their findings indicate that water resources and urban living areas significantly influence precipitation complexity, exhibiting a negative correlation.

In this work we investigate whether climate change affects the complexity of precipitation in Serbia, by applying recently

introduced Generalized Weighted Permutation Entropy method (GWPE) (Stosic et al., 2022), that enables to evaluate the complexity of temporal fluctuations over a wide range of scales, shedding new light on the underlying process. Previous studies related to temporal changes in precipitation in Serbia were based on trend analysis (Milovanović et al., 2017; Luković et al., 2014), extreme indices (Malinović-Milićević et al., 2016; Tošić et al., 2025) and changes in precipitation seasonality (Stosic





et al., 2024a; Amiri and Gocić, 2025). Recently climate variability in Serbia was addressed by using the concept of complexity.

Mimić et al. (2017) applied two complexity measures, Kolmogorov entropy and Sample entropy, on the daily maximum and minimum air temperature and the precipitation time series from seven stations in Serbia from the period 1951–2010. They found that both Kolmogorov complexity and Sample entropy of the maximum temperature series showed increasing trends for all stations, but it was statistically significant only for Kolmogorov entropy. The trends of complexity measures for the minimum temperature depend on the location, but without statistical significance except for locations in eastern (Negotin) and

southern (Vranje) Serbia, where positive trend was found for Kolmogorov entropy and Sample entropy, respectively. For precipitation series both complexity measures showed decreasing trend which was statistically significant only for the Negotin station.

We address the influence of climate change on the complexity of rainfall dynamics in Serbia trough the GWPE analysis of daily precipitation temporal series from 14 synoptic stations. We compare the GWPE curves in complexity/entropy causality plane for two 30-year sub-periods: 1961-1990 and 1991-2020. The main objective of this work is threefold, to find i) if

complexity of temporal fluctuations of precipitation in Serbia varies over a wide range of scales; ii) if these properties vary spatially; iii) if precipitation complexity is affected by climate change.

## 2 Data and methodology

### 2.1 Study area and dataset

Serbia is a continental country located in the western part of the Balkan peninsula in the southeast of Europe, between latitudes 41°50′ and 46°10′ N (Fig. 1). The northern part is mostly flat with low elevation terrain, which gradually turns into hills and mountains that surround river valleys as moving towards central and southern part. In the northern part the climate type is moderate continental, changing to continental climate in the central part, and modified Mediterranean climate in the southern and the southwestern part (Bajat et al., 2015). In the lowlands the mean annual temperature is between 11 and 12 °C and mean

annual precipitation between 500 and 700 mm, while in the mountains, the mean annual temperature is below 8 °C, and the mean annual precipitation is above 1000 mm (Vujadinović Mandić et al., 2022).

The data used in this work is the daily precipitation amount recorded during the period 1961-2020 at 14 stations located across Serbia, as shown in Fig. 1. The chosen stations have all the measurements during the study period. The data are provided by the Serbian Meteorological Service, which performed technical and critical controls of these measurements. The geographic

information on the 14 stations is given in Table 1.





**Table 1: Stations with their geographic information (latitudes, longitudes and altitudes).**

| Abbreviations | Stations | Latitude | Longitude | Altitude (m) |
|---|---|---|---|---|
| BG | Belgrade | 44°48' N | 20°28' E | 132 |
| KG | Kragujevac | 44°02' N | 20°56' E | 185 |
| KR | Kraljevo | 43°44' N | 20°41' E | 215 |
| LO | Loznica | 44°33' N | 19°14' E | 121 |
| NE | Negotin | 44°13' N | 22°31' E | 42 |
| NI | Niš | 43°20' N | 21°54' E | 201 |
| NS | Novi Sad | 45°20' N | 19°51' E | 84 |
| SP | Smederevska Palanka | 44°22' N | 20°57' E | 122 |
| SO | Sombor | 45°47' N | 19°05' E | 88 |
| SR | Sremska Mitrovica | 44°58' N | 19°38' E | 81 |
| VG | Veliko Gradište | 44°45' N | 21°31' E | 82 |
| ZA | Zaječar | 43°53' N | 22°17' E | 144 |
| ZL | Zlatibor | 43°39' N | 19°41' E | 1085 |
| ZR | Zrenjanin | 45°24' N | 20°21' E | 80 |

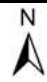

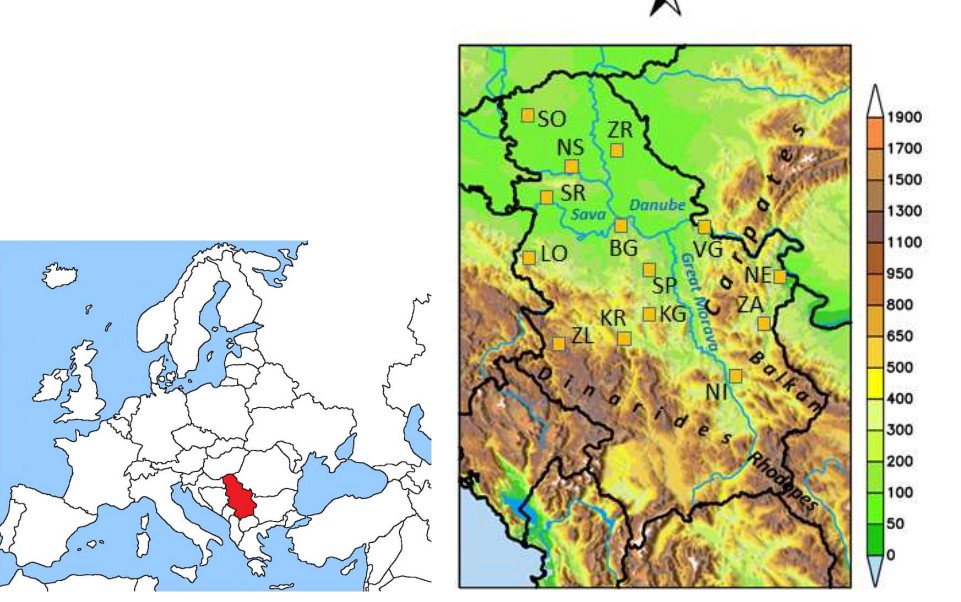

**Figure 1: Location of Serbia in Europe (left, https://commons.wikimedia.org/wiki/File:Location_map_of_Serbia_in_Europe_%282006%E2%80%932008%29.gif) and orography of Serbia with locations of meteorological stations (right).**



## 2.2 Generalized weighted permutation entropy

The Generalized Weighted Permutation Entropy (GWPE) method was recently introduced (Stosic et al., 2022) as the
generalization of Permutation entropy (PE) (Bandt and Pompe, 2002) and Weighted permutation entropy method (WPE)
(Fadlallah et al., 2013). While PE and WPE evaluate overall predictability of time series, GWPE is capable of distinguishing
among the predictability of subsets of small/large fluctuations and thus provides a deeper insight into the time series behavior.
Along with PE, WPE and GWPE, statistical complexity can be defined, and measures of entropy and complexity can be
represented and jointly analyzed in the Complexity entropy causality plane (CECP) (Rosso et al., 2007).

In the PE method (Bandt and Pompe, 2002), first overlapping segments (words) of a given size $d > 1$ are extracted along the
time series $x_t, t = 1, \dots, N$, denoted as $X_s = (x_s, x_s, \dots, x_{s+d-1}), s = 1, \dots, N - (d - 1)$. The values within each segment are
sorted in increasing order $x_{s+r_0} \le x_{s+r_1} \le \cdots \le x_{s+r_{d-1}}$, forming the vector of indices $v_s \equiv (r_0, r_1, \dots, r_{d-1})$, which represent
one of the $d!$ possible permutations of integers $0, 1, \dots, d - 1$, symbolically encoding the original segment. The relative
frequencies of these permutations $\pi_i, i = 1, \dots, d!$, form the empirical distribution $P \equiv \{p(\pi_i), i = 1, \dots, d!\}$, which is used
to calculate the permutation entropy as

$$H(P) = -\sum_{i=1}^{d!} p(\pi_i) \log p(\pi_i), \tag{1}$$

To accurately estimate $H(P)$, the length of the time series should be considerably greater than $d!$ For real-world data, selecting
the maximum embedding dimension that satisfies $T > 5d!$ is shown to be sufficient to get good statistics (Riedl et al., 2013).
In the CECP method (Rosso et al., 2007) the Jensen-Shannon divergence

$$J(P, U) = \left\{ H\left[\frac{P+U}{2}\right] - H\left[\frac{P}{2}\right] - H\left[\frac{U}{2}\right] \right\}, \tag{2}$$

which measures the distance of the empirical pattern distribution from a uniform distribution, is used to calculate the
normalized complexity measure

$$C(P) = \frac{H(P)}{d!} \frac{J(P,U)}{J_{max}}, \tag{3}$$

where $J_{max}$ is the maximum possible value of $J(P, U)$

$$J_{max} = -\frac{1}{2}\left[\frac{d!+1}{d!}\log(d! + 1) - 2\log 2d! + \log d!\right] \tag{4}$$

obtained when only a single pattern is observed. The CECP represents a two-dimensional plot where $H(P)$ is used for the
horizontal and $C(P)$ for the vertical coordinate.

The Weighted Permutation Entropy (WPE) (Fadlallah et al., 2013) introduced to account for the amplitude of the values in the
segments. Instead of using relative pattern frequencies, WPE calculates probability as

$$p(\pi_i) = \frac{\sum_{s, \pi_s = \pi_i} w_s}{\sum_s w_s}, \tag{5}$$



where $w_s = 1/d \sum_{i=0}^{d-1}(x_{s+1} - \langle x_s \rangle)^2$ is the variance, and $\langle x_s \rangle = 1/d \sum_{i=0}^{d-1} x_{s+1}$ is the average of the values observed in segment. The WPE and the corresponding CECP extension (WCECP) are then implemented using the above Eqs. (1-4)

These methods have proven valuable, with applications spanning fields such as medicine (Deng et al., 2015), finance (Zunino et al., 2010), ecology (Sippel et al., 2016), geophysics (Konstantinou et al., 2022), and hydrology (de Carvalho Barreto et al., 2023). Additionally, some studies (Xiaet al., 2016) have demonstrated that the weighted variants outperform the original methods.

The Generalized Weighted Permutation Entropy (GWPE) method has been recently proposed (Stosic et al., 2022) as an extension of the earlier approaches, incorporating both PE and WPE as particular cases. It is achieved through the probability definition

$$p(\pi_i, q) = \frac{\sum'_{s, \pi_s = \pi_i} w_s^{q/2}}{\sum'_s w_s^{q/2}}, \tag{6}$$

where $-\infty < q < \infty$ is a continuous scaling parameter. The prime in the summation indicates that the segments with strictly zero variance $w_s$ are omitted. The scaling parameter $q$ acts as a "magnifying glass", where negative q values emphasize small fluctuations, and positive q values enhance large fluctuations. The PE method corresponds to GWPE for $q = 0$, and WPE corresponds to GWPE for $q = 2$. Although it was only recently introduced, GWPE has already found applications in data analysis in several studies (Stosic et al., 2024b; Stosic and Stosic, 2024; Duarte et al., 2025). Moreover, it was shown in ref. (Duarte et al., 2025) that GWPE outperformed PE and WPE in distinguishing between different sleep stages from EEG signals.

## 3 Results and discussion

Mean annual precipitation for two subperiods for all 14 stations and the change from the first to second period is presented in Fig. 2, while spatial distribution of these values across Serbia is shown in Fig. 3. In both periods annual precipitation varies between 550 mm and 1050 mm. In the first period (1961-1990) the lowest precipitation (550-650 mm) was observed in the northern lowland part (Sombor, Zrenjanin, Novi Sad, Sremska Mitrovica), in valleys along the rivers Velika Morava in the central part (Smederevska Palanka, Kragujevac), Južna Morava in the southern part (Niš) and in the eastern part (Negotin and Zaječar). The intermediate precipitation (650-750 mm) was observed in the region along the Danube River (Belgrade, Veliko Gradište) and in Zapadna Morava valley (Kraljevo), while the highest precipitation (800-1000 mm) was observed in the western mountainous region (Loznica, Zlatibor). The lowest average precipitation was observed in Zrenjanin (555.8 mm) and the highest in Zlatibor (964.1 mm). In the second period (1991-2020) the precipitation increased in most of Serbian territory with largest increase in Novi Sad (151 mm). In Sremska Mitrovica, Veliko Gradište, Negotin, Zaječar and Kraljevo the mean annual precipitation did not change in the second period. The lowest average precipitation was observed in Zrenjanun (597.2 mm) and the highest in Zlatibor (1030.4 mm).




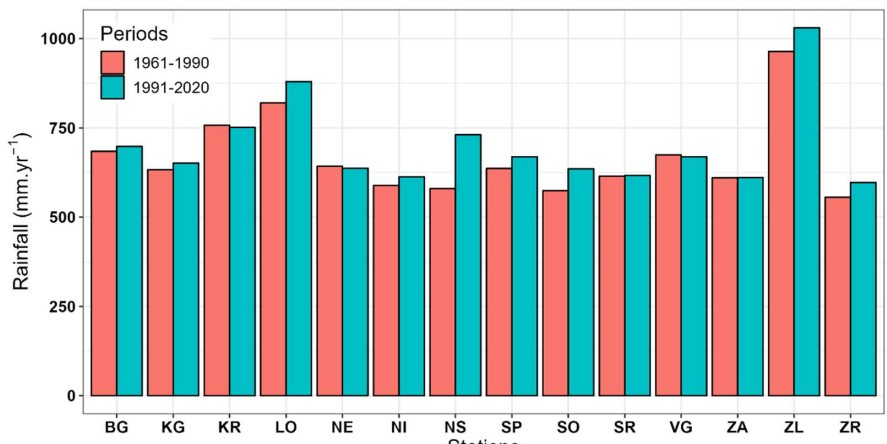

**Figure 2: Mean annual precipitation for two periods (1961-1990 and 1991-2020) for 14 stations in Serbia.**

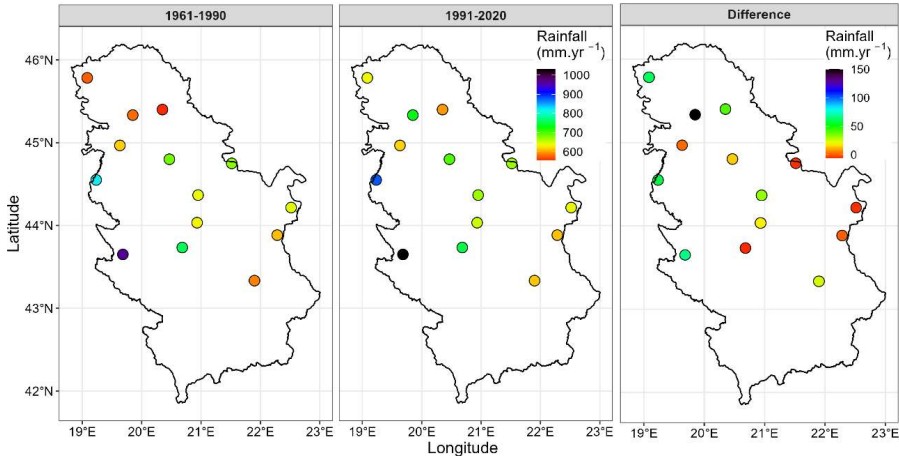

**Figure 3: Spatial distribution of mean annual precipitation in Serbia, for two periods: (1961-1990 and 1991-2020) and change from first to second period (Difference).**

The GWPE method was applied on precipitation anomalies (deseasonalized) series $y_{i,j} = (x_{i,j} - \mu_i)/\sigma_i$ where $x_{i,j}$ is the precipitation value recorded on a given calendar day $i$ of year $j$, $\mu_i$ and $\sigma_i$ are mean and standard deviation calculated for each calendar day $i$ over the years $j$ of the observation period. GWPE curves in two-dimensional entropy complexity plane for all stations and for two considered sub-periods are presented in Fig. 4 and Fig. 5, respectively. In both periods and across all locations, entropy values are consistently lower for $q < 0$, indicating higher predictability of small fluctuations.





### Belgrade

### Kragujevac

### Kraljevo

### Loznica

### Negotin

### Niš

### Novi Sad

### Smederevska Palanka

### Sombor

### Sremska Mitrovica

### Veliko Gradište

### Zaječar

### Zlatibor

### Zrenjanin



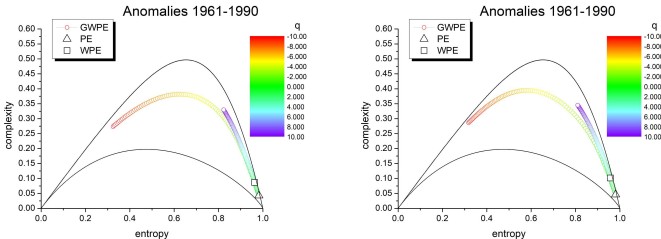

**Figure 4: GWPE curve (entropy/complexity as a function of magnification parameter q) for 14 stations in Serbia for the first sub-period (1961-1990). The full lines represent the theoretical bounds for segment size of w=6 days (Martin et al., 2006).**

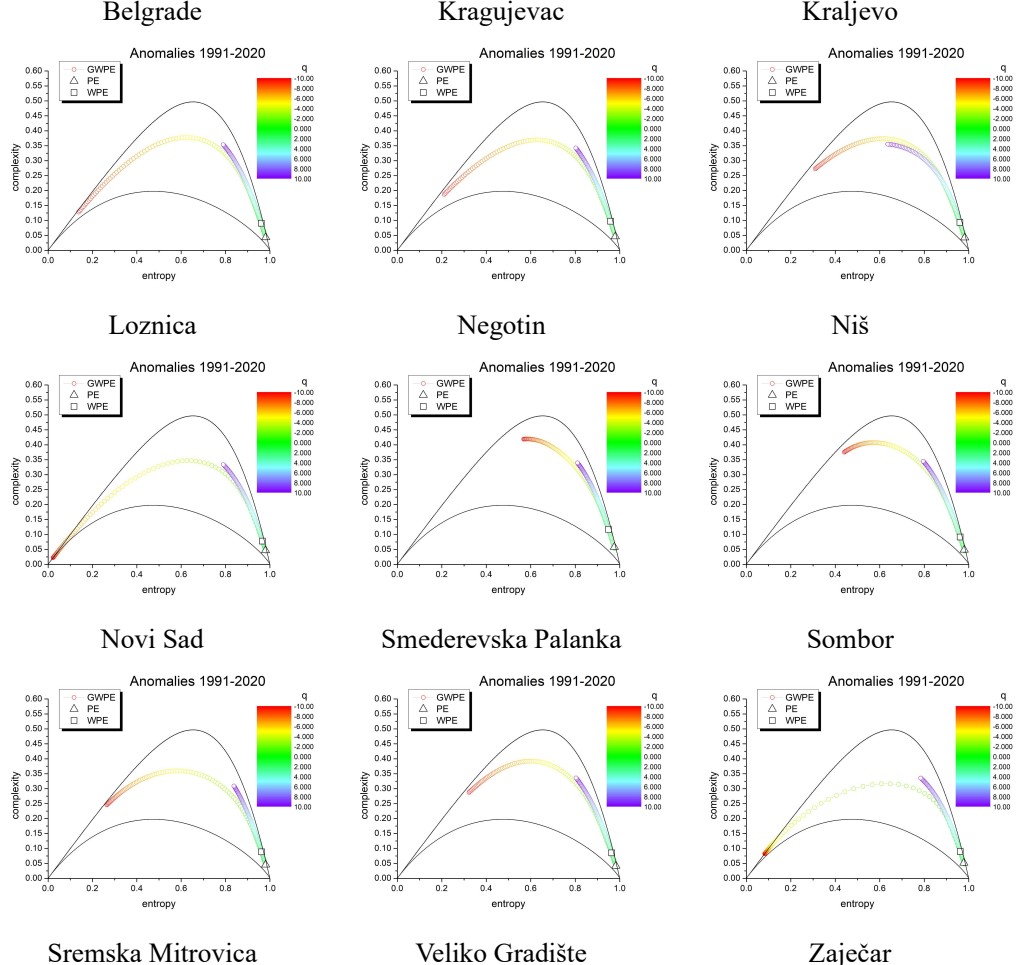





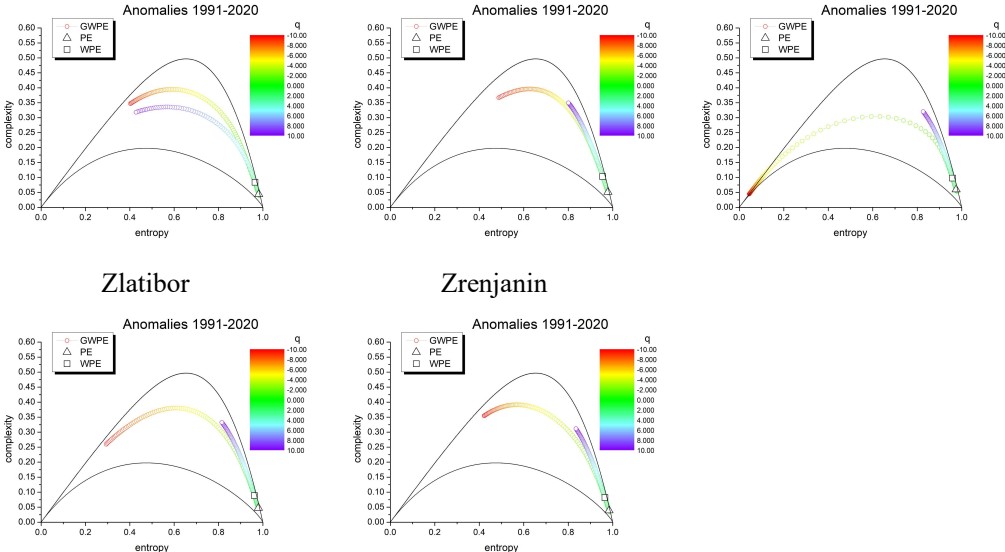

Zlatibor                Zrenjanin

**Figure 5: GWPE curve (entropy/complexity as a function of magnification parameter q) for 14 stations in Serbia for the second sub-period (1991-2020). The full lines represent the theoretical bounds for segment size of w=6 days (Martin et al., 2006).**

A notable exception is observed in Sremska Mitrovica during the second period, where the curve that describes behavior of small fluctuations ($q < 0$) lies below that for large fluctuations ($q > 0$). This implies that, for subsets of fluctuations with identical entropy values, the larger fluctuations exhibit greater complexity. In all cases, Weighted permutation entropy (WPE) values are lower than their corresponding Permutation entropy (PE) values, while associated complexity values are comparatively higher. Considering statistical complexity, it is observed that for $q > 0$, the points are located in the right side of the GWCECP plane, where the decrease of entropy is followed by the increase of statistical complexity, while for $q < 0$ points are mostly located in the left side of CWCECP plane where both entropy and statistical complexity simultaneously decrease/increase. These observations are further supported by the complexity-entropy-scale causality boxes (CESCB) (Stosic et al., 2022), which plot entropy, complexity, and magnification factor $q$ as coordinates in the three-dimensional space, as depicted in Fig. 6 and Fig. 7.





### Belgrade

Anomalies 1991-2020

### Kragujevac

Anomalies 1991-2020

### Kraljevo

Anomalies 1991-2020

### Loznica

Anomalies 1991-2020

### Negotin

Anomalies 1991-2020

### Niš

Anomalies 1991-2020

### Novi Sad

Anomalies 1991-2020

### Smederevska Palanka

Anomalies 1991-2020

### Sombor

Anomalies 1991-2020

### Sremska Mitrovica

Anomalies 1991-2020

### Veliko Gradište

Anomalies 1991-2020

### Zaječar

Anomalies 1991-2020

### Zlatibor

### Zrenjanin




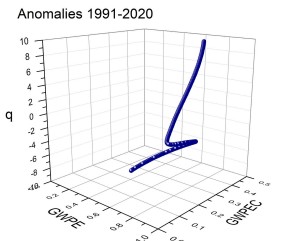
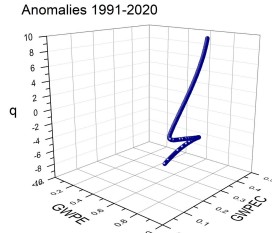

**Figure 6: Three-dimensional GWPE causality box for 14 stations in Serbia for the first sub-period (1961-1990).**

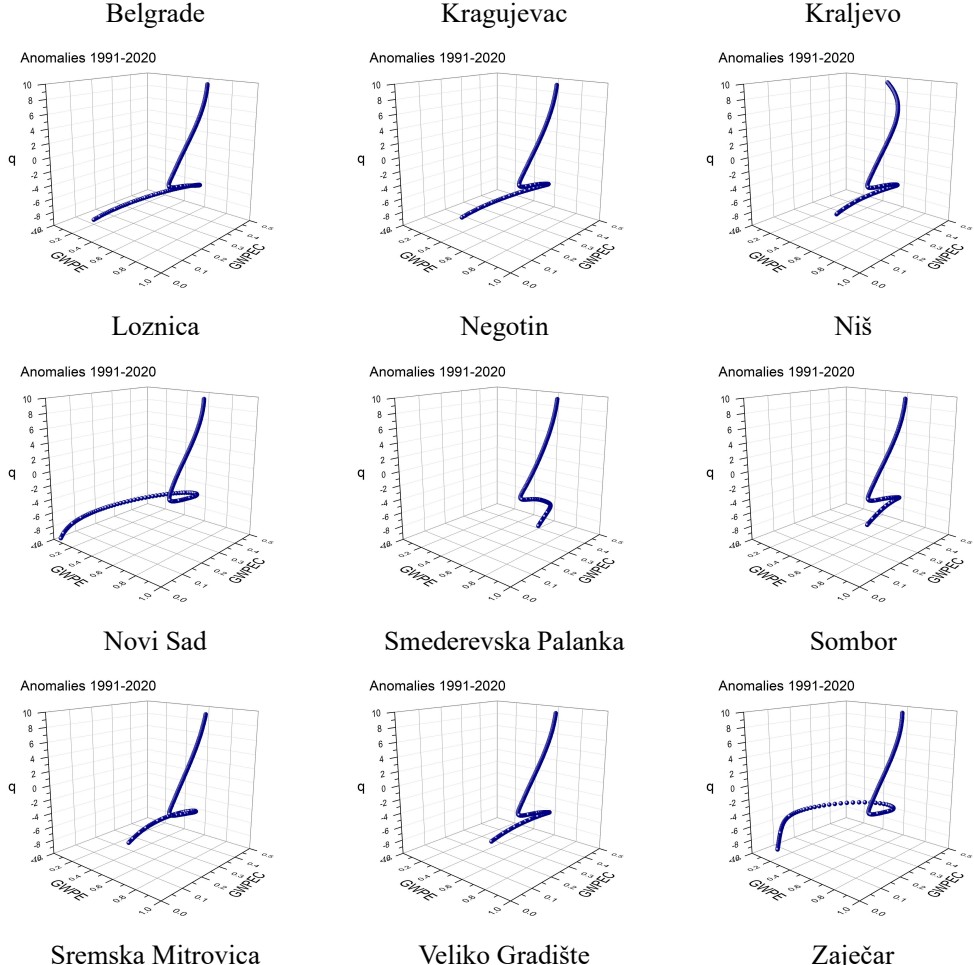

Belgrade

Kragujevac

Kraljevo

Loznica

Negotin

Niš

Novi Sad

Smederevska Palanka

Sombor

Sremska Mitrovica

Veliko Gradište

Zaječar





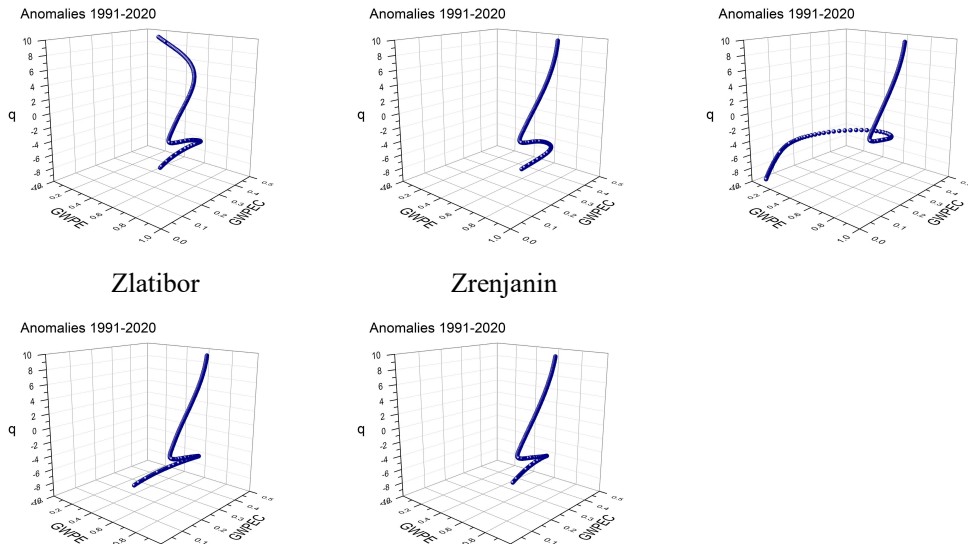

Zlatibor          Zrenjanin

**Figure 7: Three-dimensional GWPE causality box for 14 stations in Serbia for the second sub-period (1991-2020).**

The values of GWPE and GWPEC for $q = -10$, $q = 10$, $q = 0$, and $q = 2$ for all stations and for two considered sub-periods are presented in Table 2 and Table 3, respectively. It is seen that GWPE and GWPEC values for $q = 0$ and $q = 2$ which correspond to the PE and WPE methods respectively, remain relatively stable across stations and time intervals. In contrast, the values of entropy (GWPE) and statistical complexity (GWPEC) for $q = -10$ and $q = 10$ exhibit noticeable variation both spatially and temporally between the two 30 years periods.

For enhanced interpretability, the GWPE and GWPEC results for all stations and both sub-periods and the change from the first to second period are shown in Fig. 8 and Fig. 9 respectively. It is seen from Table 2 and Fig. 8 that GWPE values at $q = -10$ are consistently lower across both periods, indicating greater predictability of small fluctuations of precipitation.

In the first sub-period small fluctuations ($q = -10$) exhibited the highest predictability (lowest GWPE values) in Sombor (0.012), Zaječar (0.014), Loznica (0.022), and Kragujevac (0.072), while the lowest predictability (highest GWPE values) was observed in Smederevska Palanka (0.419), and Niš (0.383). Between the first and second sub-periods, entropy values, and consequently, predictability of rainfall fluctuations changed across stations. An increase in entropy (reflecting decreased predictability) was noted in Kragujevac, Negotin, Niš, Sremska Mitrovica, Veliko Gradište, and Zrenjanin. Conversely, predictability improved (entropy decreased) at Belgrade, Kraljevo, Smederevska Palanka, and Zlatibor. The remaining stations Novi Sad, Sombor, Zaječar, and Loznica, showed no change in entropy values. Among all stations, Belgrade exhibited the most substantial gain in predictability, with entropy decreasing from 0.360 to 0.138, while Negotin recorded the most significant loss, with entropy rising from 0.138 to 0.360. At these locations, although the average annual precipitation remained stable in the second sub-period, significant changes were observed in the predictability of small fluctuations, suggesting that



even in the absence of mean precipitation shifts, underlying complexity and short-term variability can evolve markedly. In
contrast, at Novi Sad, where the most pronounced increase in precipitation occurred, predictability of small fluctuations
remained unchanged, indicating a decoupling between precipitation volume and temporal structure.

The predictability of small rainfall fluctuations is crucial for estimation of hydrological impacts such as surface moisture
retention and evapotranspiration dynamics (Yan et al., 2021; Cui et al., 2022). It is also important for agricultural planning as
predictable light rainfall helps optimize irrigation and planting time. Prolonged low intensity rainfall can trigger urban
landslides and significantly contribute to soil erosion (Dunkerley, 2021). Spatial patterns reveal an increase in the predictability
(lower entropy values) of small precipitation fluctuations across northern and eastern Serbia, which could be associated with
increasing local atmospheric order. In contrast, the central and western regions show a decline in predictability suggesting
increased climatic irregularity or external forcing mechanisms that elevate entropy (Fig. 8a).

For all stations and both sub-periods, entropy values at $q = 10$ are consistently higher than those at $q = -10$, indicating that
large precipitation fluctuations are less predictable than small ones. This reduced predictability poses challenges for
agricultural planning, water resource management, and the implementation of mitigation strategies for hydrological extremes
such as floods and droughts (Sultan et al., 2005; Butterworth et al., 1999; Zhou et al., 2023).

Table 2: The values of GWPE for $q=-10$, $q= 0$, and $q=2$, $q=10$, for14 stations in Serbia for the first sub-period (1961-1990).

| City | 1961-1990 | | | | 1991-2020 | | | |
|---|---|---|---|---|---|---|---|---|
| | $q=-10$ | $q=0$ | $q=2$ | $q=10$ | $q=-10$ | $q=0$ | $q=2$ | $q=10$ |
| Belgrade | 0.360 | 0.982 | 0.961 | 0.787 | 0.138 | 0.981 | 0.961 | 0.790 |
| Kragujevac | 0.072 | 0.980 | 0.958 | 0.813 | 0.211 | 0.979 | 0.959 | 0.803 |
| Kraljevo | 0.355 | 0.983 | 0.959 | 0.765 | 0.310 | 0.982 | 0.960 | 0.636 |
| Loznica | 0.022 | 0.981 | 0.958 | 0.747 | 0.023 | 0.980 | 0.967 | 0.790 |
| Negotin | 0.237 | 0.978 | 0.949 | 0.756 | 0.568 | 0.976 | 0.950 | 0.810 |
| Nis | 0.383 | 0.981 | 0.956 | 0.812 | 0.441 | 0.980 | 0.962 | 0.795 |
| Novi Sad | 0.257 | 0.982 | 0.961 | 0.783 | 0.265 | 0.981 | 0.963 | 0.838 |
| Smederevska Palanka | 0.419 | 0.982 | 0.960 | 0.773 | 0.323 | 0.983 | 0.964 | 0.804 |
| Sombor | 0.012 | 0.979 | 0.966 | 0.777 | 0.083 | 0.978 | 0.962 | 0.785 |
| Sremska Mitrovica | 0.331 | 0.982 | 0.961 | 0.780 | 0.403 | 0.981 | 0.965 | 0.428 |
| Veliko Gradište | 0.359 | 0.981 | 0.959 | 0.783 | 0.488 | 0.979 | 0.955 | 0.801 |
| Zaječar | 0.014 | 0.978 | 0.953 | 0.790 | 0.045 | 0.975 | 0.959 | 0.826 |
| Zlatibor | 0.325 | 0.982 | 0.963 | 0.823 | 0.293 | 0.980 | 0.962 | 0.815 |
| Zrenjanin | 0.318 | 0.981 | 0.957 | 0.810 | 0.422 | 0.984 | 0.965 | 0.835 |


While entropy values at $q = 10$ show limited spatial variation across stations (Fig. 8b) there are evident temporal changes
between the two sub-periods (Table 2). A slight increase in entropy values was observed at seven stations, predominantly in
the eastern (Negotin, Veliko Gradište, Zaječar) and northern (Novi Sad, Zrenjanin) regions suggesting a decline in the
predictability of large fluctuations, while three stations exhibited a slight entropy reduction, indicating improved predictability.
The most pronounced changes occurred at Kraljevo (central region) where entropy decreased from 0.765 to 0.636, and Sremska





Mitrovica (northern region) which showed a substantial decrease from 0.780 to 0.428. These results indicate an improvement
in the predictability of large precipitation fluctuations at these locations during the second sub-period, despite the stability in
average annual precipitation. In contrast, Zlatibor and Sombor experienced an increase in precipitation over the same period,
yet the predictability of large fluctuations remained unchanged, highlighting once again a decoupling between precipitation
volume and its temporal dynamics.

**Table 3: The values of GWPEC for $q$=-10, $q$= 0, and $q$=2, $q$=10, for14 stations in Serbia for the second sub-period (1991-2020).**

| | 1961-1990 | | | | 1991-2020 | | | |
|---|---|---|---|---|---|---|---|---|
| City | $q$=-10 | $q$=0 | $q$=2 | $q$=10 | $q$=-10 | $q$=0 | $q$=2 | $q$=10 |
| Belgrade | 0.314 | 0.042 | 0.091 | 0.350 | 0.128 | 0.044 | 0.091 | 0.354 |
| Kragujevac | 0.070 | 0.047 | 0.099 | 0.338 | 0.187 | 0.049 | 0.097 | 0.343 |
| Kraljevo | 0.319 | 0.040 | 0.095 | 0.364 | 0.273 | 0.043 | 0.093 | 0.355 |
| Loznica | 0.022 | 0.046 | 0.100 | 0.371 | 0.022 | 0.047 | 0.078 | 0.334 |
| Negotin | 0.210 | 0.052 | 0.118 | 0.368 | 0.419 | 0.058 | 0.117 | 0.339 |
| Nis | 0.336 | 0.047 | 0.102 | 0.340 | 0.375 | 0.049 | 0.091 | 0.345 |
| Novi Sad | 0.235 | 0.044 | 0.093 | 0.362 | 0.245 | 0.046 | 0.089 | 0.309 |
| Smederevska Palanka | 0.347 | 0.043 | 0.094 | 0.360 | 0.287 | 0.042 | 0.085 | 0.336 |
| Sombor | 0.012 | 0.049 | 0.080 | 0.347 | 0.082 | 0.052 | 0.090 | 0.336 |
| Sremska Mitrovica | 0.286 | 0.044 | 0.093 | 0.347 | 0.347 | 0.044 | 0.084 | 0.318 |
| Veliko Gradište | 0.292 | 0.045 | 0.095 | 0.350 | 0.367 | 0.051 | 0.103 | 0.350 |
| Zaječar | 0.014 | 0.053 | 0.109 | 0.366 | 0.045 | 0.059 | 0.098 | 0.320 |
| Zlatibor | 0.274 | 0.043 | 0.086 | 0.330 | 0.260 | 0.048 | 0.089 | 0.332 |
| Zrenjanin | 0.287 | 0.046 | 0.101 | 0.345 | 0.354 | 0.039 | 0.082 | 0.312 |




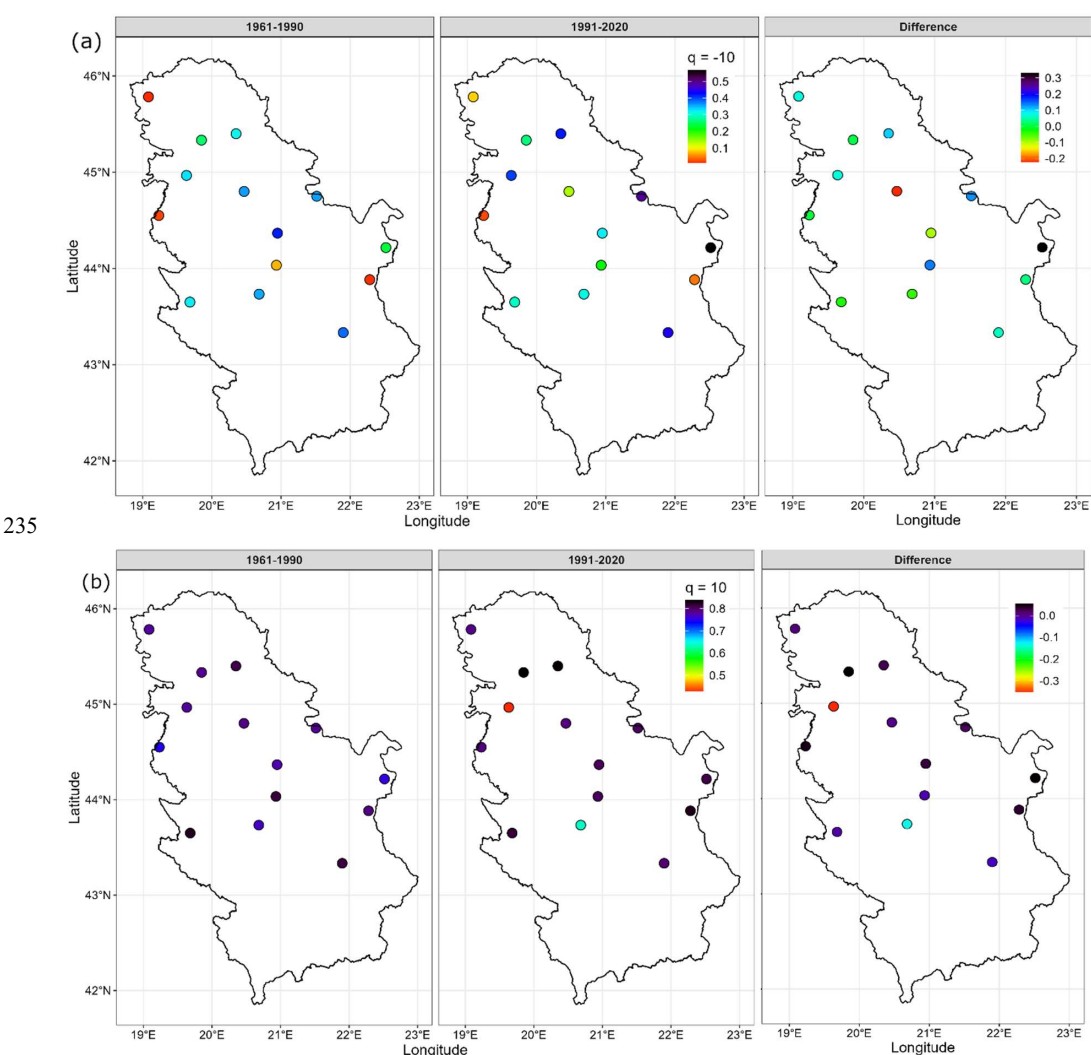

**Figure 8: GWPE for *q* = -10 (a) and *q* = 10 (b) for 14 stations in Serbia for two periods (1961-1990 and 1991-2020) and change from first to second period.**



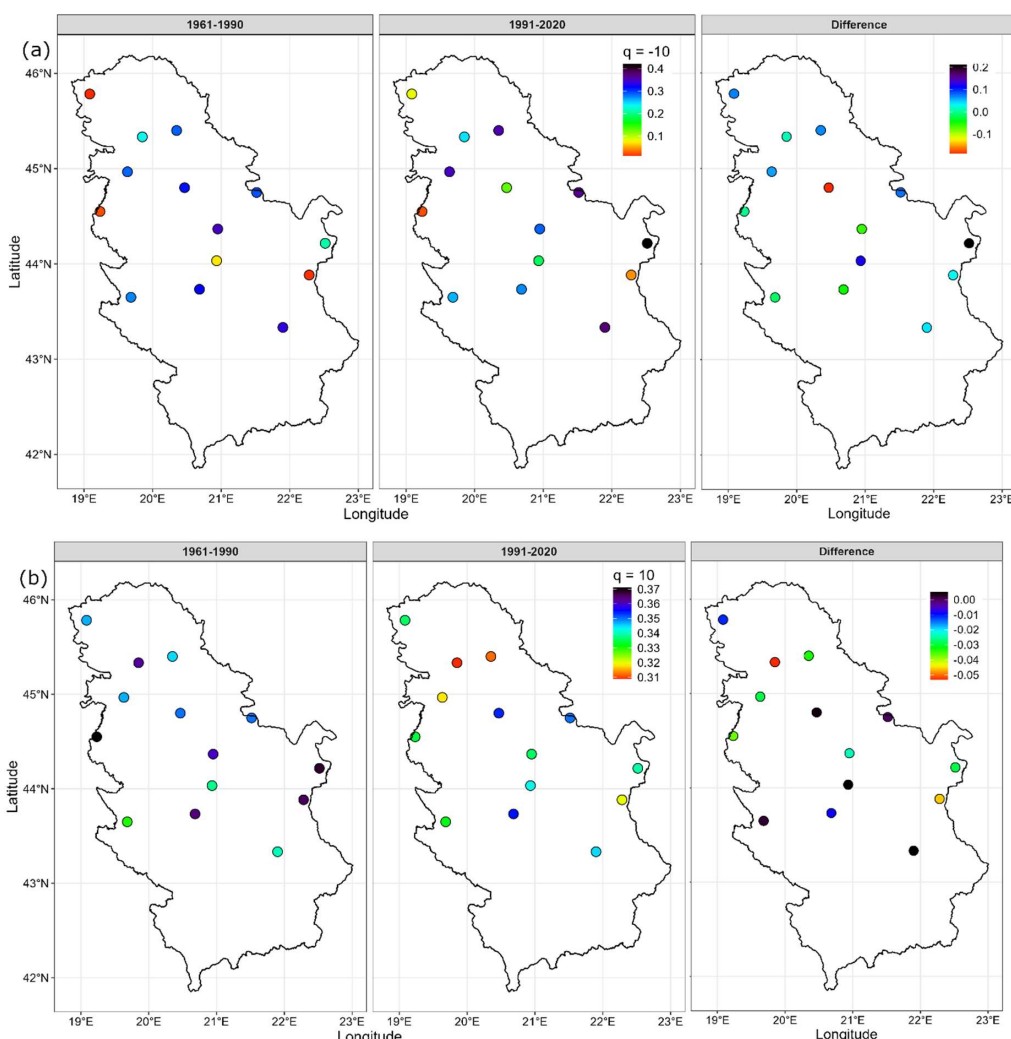


**Figure 9: GWPEC for *q* = -10 (a) and *q* = 10 (b) for14 stations in Serbia for two periods (1961-1990 and 1991-2020) and change from first to second period (Difference).**

## 4 Conclusions

Understanding of temporal and spatial patterns of precipitation and their alterations under the changing climate is crucial for climate impact studies. While statistical methods remain the primary tools for evaluation of these changes, recent findings





suggest that emergent properties arising from complex interactions among processes that govern climate variability, may also be influenced by climate change.

The objective of this study was to examine whether one of such emergent property, complexity, is characteristic of precipitation time series in Serbia, and whether it has been affected by climate change. To this end, we applied the Generalized Weighted Permutation Entropy (GWPE) method to precipitation records from 14 weather stations distributed cross Serbia, spanning the period 1961-2020.

The main results of this work are: i) GWPE values for $q = 0$ and $q = 2$ which corresponding to Permutation entropy (PE) and Weighted permutation entropy (WPE) respectively, remain relatively stable across stations and time intervals; ii) GWPE values describing the predictability of small ($q = -10$) and large ($q = 10$) precipitation fluctuations exhibited noticeable spatial and temporal variation between the two 30-year periods; iii) GWPE values at $q = -10$ were consistently lower across both periods, indicating greater predictability of small precipitation fluctuations. In some cases, significant changes in entropy values occurred despite stable annual precipitation totals. Conversely, in other locations, entropy values remained steady even as annual totals varied, suggesting the decoupling between precipitation volume and its underlying temporal structure. These findings suggest that the complexity of precipitation dynamics in Serbia (measured via GWPE) has been influenced by climate change.

The GWPE shows strong potential to capture hidden variability in precipitation behavior and detect climate-related impacts that may be masked if focusing only on traditional statistical averages. Future work should compare these results with outputs from climate models under various emission scenarios, complementing classical statistical analyses to help identify the most suitable model or ensemble for climate change studies in this region.

**Code and Data availability**

The datasets generated and/or analysed during the current study are available from the corresponding author upon reasonable request.

**Author contribution**

The conception of this study was the responsibility of TS. Material preparation, data collection, and analysis were performed by TS, IT, ASAS, VDj and BS. The paper was written by TS and revised by BS. All the authors have read and approved the submission of paper.

**Competing interests**

The authors declare that they have no conflict of interest.



**Special issue statement**

This article is part of the special issue "Emerging predictability, prediction, and early-warning approaches in climate science".

**Acknowledgments**

TS and BS acknowledge support of Brazilian agency CNPq (grants No 308782/2022-4 and 309499/2022-4). B.S. acknowledges support of Brazilian agency CAPES through grant No 88887.937789/2024-00. I.T. and V.D. acknowledge the
support of the Ministry of Science, Technological Development and Innovation of the Republic of Serbia, grant No. 451-03-136/2025-03/200162.

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
