# Peer review of "Spatiotemporal variation in rainfall predictability in Serbia under a changing climate"

_EGUsphere, 2025_

## Author Comment (AC1)

We thank the Reviewers for constructive criticism that has led to an improvement of this manuscript. In what follows we present the responses (highlighted in blue) to Reviewers' concerns.

**Reviewer #1**

I find this manuscript and its findings both interesting and valuable for advancing our understanding of the complexity and predictability of rainfall records, particularly in light of the current uncertainty surrounding general rainfall predictions. The methods used for data analysis are appropriate and reproducible, and I especially commend the application of the relatively new entropy-based calculation method. Overall, I recommend this manuscript for publication in EGUsphere.

My questions and comments are as follows:

1.  The use of two climatic time periods for separate analyses is clear. However, I am also interested in whether analyzing the entire available period (1961–2020) would reveal visible changes in the GWPE plots (e.g., crossovers) that might indicate shifts in predictability between the two climatic periods. Could you provide results or insights for the whole period?

    Reply: If GWPE is calculated using 30-year sliding windows (the standard climatological period), this would yield temporal series of GWPE values for each selected value of q (e.g., q = -10, 0, 2, 10). Such an approach would allow for the identification of potential crossovers and long-term trends in predictability. In future work, we intend to apply this analysis using a longer dataset to better capture temporal dynamics and enhance interpretability.

2.  The GWPE values are presented with three significant digits. Does this imply that the error of calculation lies in the third digit? More generally, is there a way to estimate or quantify the error associated with this method?

    Reply: While this is a scientifically relevant question, it remains unaddressed in the current body of literature: the choice of three significant digits is a matter of convention.

3.  In Figures 3, 8, and 9, would it be possible to display the differences on a color scale distinct from those used for rainfall amount or GWPEC in the two periods? Similarly, in Tables 2 and 3, could you add columns showing the differences in values between the two periods?

    Reply: Done, thank you.

4.  For q>0, GWPE values all appear higher than 0.5, except for Sremska Mitrovica during the 1991–2020 period. This suggests a possible pattern—has a similar behavior been observed in other types of real-world data?

    Reply: GWPE is a newly developed method, and this study represents its first application to precipitation data. Previous applications include hot pixel data from Brazilian biomes (Stosic and Stosic, 2024), SPEI time series in Serbia (Stosic et al., 2024b.), and EEG signals across different sleep stages (Duarte et al., 2025). In all these cases, GWPE values exceeded 0.5 for both positive and negative q values. Future studies using precipitation datasets are needed

to determine whether GWPE values greater than 0.5 for q > 0 represent an intrinsic characteristic of precipitation dynamics.

Thank you for your responses and for the valuable contribution to this special issue.

**Reply:** Thank you.

---

## Author Comment (AC2)

We thank the Reviewers for constructive criticism that has led to an improvement of this manuscript. In what follows we present the responses (highlighted in blue) to Reviewers' concerns.

**Reviewer #2**

This manuscript offers an interesting and original analysis of precipitation predictability in Serbia under climate change. By combining PE, WPE, GWPE, and statistical complexity across different fluctuation scales, the study provides a novel perspective and yields insightful results that deepen our understanding of precipitation's statistical predictability. I recommend publication after the authors address several important issues noted below.

1. The authors have carried out a detailed and multi-dimensional analysis. However, the discussion mainly focuses on describing the results, with less attention to the underlying causes. It would strengthen the paper if the authors could further elaborate on the possible drivers behind the observed changes in predictability between the two periods and discuss the reasons for the spatial distribution patterns. Since only 14 sites are included, it might also be useful to include the discussion of the resulting uncertainty.

   **Reply:** We have now added the text:

   Precipitation variability in Serbia is largely driven by atmospheric circulation patterns, notably the Arctic Oscillation (AO), East Atlantic/Western Russia Oscillation (EA/WR), and North Atlantic Oscillation (NAO) (Pavlović-Berdon, 2013; Tošić, 2004, 2014; Milošević, 2021). Ruml et al. (2016) examined spatiotemporal variations in temperature and precipitation across Serbia using observational data from 26 meteorological stations spanning the period 1961–2010. Their findings indicated that annual precipitation did not exhibit significant changes over the full study period. However, during the sub-period 1986–2010, (which overlaps with our second subperiod from 1991 to 2020) an increasing trend in annual precipitation was observed at numerous stations, including Belgrade, Kraljevo, and Smederevska Palanka. Notably, our analysis revealed enhanced predictability of small-scale precipitation fluctuations at these locations, suggesting a possible link between rising precipitation and improved short-term predictability.

2. The introduction offers rich background information on CECP and then introduces the GWPE method. To improve readability, it may help to clarify the conceptual or methodological link between these two frameworks, as the transition between lines 30–60 currently feels somewhat abrupt. It would be helpful for the authors to reorganize the two sections and clarify their connection and application to improve coherence.

   **Reply:** We substituted the original text:

   In this work we investigate whether climate change affects the complexity of precipitation in Serbia, by applying recently introduced Generalized Weighted Permutation Entropy method (GWPE) (Stosic et al., 2022), that enables to evaluate the complexity of temporal fluctuations over a wide range of scales shedding new light on the underlying process.

by new one:

In this study, we examine whether climate change influences the complexity of precipitation dynamics in Serbia by applying the recently introduced Generalized Weighted Permutation Entropy (GWPE) method (Stosic et al., 2022). GWPE extends the classical Permutation Entropy (PE) framework through a modification of the Bandt and Pompe probability distribution, incorporating an additional parameter $-\infty < q < \infty$. This generalization enables the evaluation of temporal complexity across multiple scales, emphasizing small fluctuations for q < 0 and large fluctuations for q > 0, thus offering novel insights into the underlying structure of precipitation variability.

3. The choice of 1990 as the dividing year between the two study periods could be explained more clearly. Would the results differ if a different division year were chosen? How sensitive are the findings to this temporal segmentation? Would analyzing the entire period as a whole yield consistent outcome?

   Reply:: Two subperiods 1961-1990 and 1991-2020 are standard climatic periods that are used to evaluate recent climate change. If GWPE is calculated using 30-year sliding windows (the standard climatological period), this would yield temporal series of GWPE values for each selected value of q (e.g., q = -10, 0, 2, 10). Such an approach would allow for the identification of potential crossovers and long-term trends in predictability. In future work, we intend to apply this analysis using a longer dataset to better capture temporal dynamics and enhance interpretability.

4. For Figure 2 (and line 145), arranging the precipitation histogram from the lowest to highest mean annual precipitation might make the result easier to interpret. Currently, the order seems somewhat random. In addition, Figure 2 shows that some stations (e.g., KR and NE) had higher precipitation in 1991–2020 than in 1961–1990, whereas the legend in Figure 3 suggests that all differences are positive—please verify whether the legend is correct. It would also be helpful if the captions for Figures 2 and 3 included more information (e.g., data sources and calculation methods). Adding station names to the map in Figure 3 could further assist in comparing with Figure 2.

   Reply: The order of meteorological stations presented in Table 1 is consistently maintained throughout all figures and in Table 2, ensuring clarity and ease of comparison across the results.

5. The font sizes and overall layout of Figures 3–8 could be adjusted to improve readability, as some text is currently quite small. The figure captions are also somewhat too brief and could include more explanatory detail—such as what each variable represents and how the figures should be interpreted—to assist readers who may be less familiar with the methodology.

   Reply: We have now extended figure captions for Fig.3 to Fig. 9, and have increased the fonts for all figures ad tables to match the text font size.

6. The meaning of the GWPEC coordinate axes in Figures 6 and 7 could be clarified. Although statistical complexity (GWPEC) is mentioned in lines 185–200, it is not defined in the

Methods section. Including a concise explanation of how this metric is calculated and interpreted would make the analysis easier to follow. Similar clarifications would be helpful for abbreviations such as CWCECP and GWCECP (line 175), which appear without prior definition or explanation.

**Reply:** We have now added a paragraph in the methodology section:

By applying formulas (2) and (3) to the choice of probability distribution (6) one obtains the Generalized Weighted Permutation Complexity (GWPEC), and combining the generalized weighted entropy and complexity the corresponding generalized weighted complexity plane (GWCECP) is obtained.

7. Finally, since several figures present similar subplots, it might be worth simplifying and refining them to improve overall clarity and visual quality. For example, combining Figures 7 and 8 might allow for a more intuitive comparison between the two time periods, and merging Figures 2 and 3 might help present the information more comprehensively

**Reply:** While we agree with the Reviewer in principle, since the figures are rather large we prefer to leave them separate.